# Domestication shaped the chromatin landscape of grain amaranth

Corbinian Graf[1], Tom S. Winkler[1,2], Peter J. Maughan[3] & Markus G. Stetter [1,2] ✉

Plant domestication has had profound impacts on the morphology and genetic diversity of crops. Beyond sequence diversity, changes in chromatin structure can play an important role in plant adaptation. However, the interplay between the chromatin landscape and plant domestication remains unclear. Here, we present a high-quality genome assembly and chromatin landscape map of the ancient pseudo-cereal, amaranth. ATAC-sequencing of multiple accessions of three grain amaranth species and two wild relatives, shows that the overall amount of accessible chromatin is highly conserved, but about 2.5% of all chromatin switched states, with a higher fraction of the genome repeatedly opening during domestication processes. These differentially accessible chromatin regions, between the crops and their wild ancestor, are species-specific and significantly associated with selective sweeps - reflecting the repeated independent domestication of amaranth. Our findings reveal the dynamic interplay between domestication and the chromatin landscape, highlighting an additional layer of diversity in crops.

Plant domestication has had profound impacts on humans, plants, and even ecosystems[1–3]. The adaptation of plant populations to the human-made environments required numerous trait changes and imposed selective pressure on plant populations. Some of the traits that changed during domestication have been studied in recent years. These include morphological[4], physiological[5], and intrinsic traits, such as metabolite composition[6] and gene expression[4].

While the impact of domestication on numerous traits and genetic diversity has been studied[7], the impact on the physical properties of DNA has only recently started to receive attention. Methylation has been shown to vary genomewide between domesticates and their wild relatives[8,9], and the change has been associated with several phenotypic changes[10–12]. In addition, genome size has changed in multiple domesticated species, even in the absence of whole genome duplications[13–15], and polyploidization has also been associated with species that harbor domesticates[16]. These examples show that the physical properties of the genome interact with selection during plant domestication. However, the extent and repeatability of such changes remain unclear.

The chromatin landscape refers to the physical arrangement of the DNA molecule within the nucleus. DNA can be tightly packed as heterochromatin (closed) or not bound to histone proteins and therefore accessible to other molecules (euchromatin)[17]. The open state of euchromatin enables transcription factors and polymerases to bind to DNA and initiate transcription. Through its regulation of DNA accessibility, the chromatin state influences gene expression, local recombination rate, transposable element (TE) activity, and genome structure[18–20]. While chromatin state changes can be plastic and vary between tissues and even between individual cells[21], heritable chromatin differences are observed on the species, population, and between-individual level[22,23].

The importance of chromatin for gene expression and other essential functions suggests that chromatin and the chromatin landscape might be involved in adaptation. For instance, the increased capability of *Saccharomyces cerevisiae* and *Dekkera bruxellensis* to ferment glucose under aerobic conditions is caused by changes in the chromatin state of mitochondrial genes, which evolved independently in both species[24]. A few studies have looked at the impact of plant domestication on chromatin accessibility. In soybean, a comparative genomic analysis revealed a 3.7% change in chromatin state[23] and approximately 21,000 chromatin loops were differentially formed between wheat (*Triticum aestivum*) and each of its wild relatives,

[1]Institute for Plant Sciences, University of Cologne, Cologne, Germany. [2]Cluster of Excellence on Plant Sciences, University of Cologne, Cologne, Germany. [3]Brigham Young University, Provo, UT, USA. ✉e-mail: m.stetter@uni-koeln.de

*Triticum durum* and *Aegilops tauschii*[25]. However, most studies to date have relied on comparative analysis between just a few accessions of the crop and the wild ancestor, limiting their power to detect chromatin state changes, since the variation can be high even between individuals of the same species[26]. Hence, a population-scale comparison along the domestication gradient should have the power to reveal the interplay between directional selection on morphological traits during domestication and the chromatin landscape.

One model that is well-suited for this purpose are the grain amaranths, *Amaranthus caudatus* L., *Amaranthus cruentus* L. and *Amaranthus hypochondriacus* L. These nutritious diploid pseudo-cereals from the Americas have been domesticated at least three times from one ancestral species (*A. hybridus*)[27]. The most commonly cultivated species, *A. hypochondriacus* and *A. cruentus*, were domesticated in Central and North America, respectively, while *A. caudatus* was domesticated in South America. The wild *Amaranthus* species, *A. quitensis*, has previously been suggested to have been involved in the domestication of *A. caudatus*[27,28]. Indeed, strong signatures of gene flow are seen between these sympatric relatives[29]. The replicated domestication and parallel selection for domestication traits make the grain amaranth species complex an ideal system to study the impact of domestication and selection on the chromatin landscape.

Here, we sequence the pan-chromatin map of 42 samples, representing the five *Amaranthus* species that are involved in the repeated domestication of the crop. We first assemble an improved reference genome, its methylome and high-resolution chromatin map through Assay for Transposase-Accessible Chromatin (ATAC) sequencing for the domesticated species *A. hypochondriacus*. We show that a number of transposon superfamilies preferentially insert into open chromatin regions, but methylation silences a large proportion of retro-transposons in open chromatin regions. While chromatin accessibility is generally conserved throughout the domestication process, a significant number of regions undergo changes in their chromatin state during domestication. Consistent with the independent domestications of the three grain amaranths, most chromatin changes are species-specific. Interestingly, several key candidate regions have overlapping selection signals, suggesting an interplay between chromatin and selection during plant domestication.

## Results

### Highly complete *A. hypochondriacus* reference genome and annotation shed light on genus evolution

To facilitate our genomic analyses, we generated an improved *A. hypochondriacus* reference genome assembly, using a sequencing depth of 30X HiFi (mean length: 13 kb) and 32x ONT (mean length: 48 kb, Supplementary Fig. 1), resulting in a total assembly length of 434,800,201 bp divided into 348 contigs (N50: 9,806,733 bp, L50: 16). The assembly was further scaffolded using Hi-C data and manually inspected to capture the 16 haploid chromosomes of *A. hypochondriacus*, producing the final 434,863,491 bp assembly consisting of 232 scaffolds (N50: 25,996,252 bp, L50: 7), including the 16 chromosome-level scaffolds, which alone comprise 96.4% of the sequence. The final assembly demonstrated high BUSCO completeness (99.3% complete BUSCOs, including 2.12% duplicated) and an assembly length increase of 31 Mb compared to the previous, primarily short-read based, reference genome (434,863,491 bp compared to 403,994,491 bp)[30,31]. The new assembly was largely collinear to the previous reference genome[30], however, chromosome 11 featured a large inversion (Supplementary Fig. 2). On chromosome 10 a potentially erroneous assembly of organellar contigs in the previous assembly[31] showed considerable rearrangement in the new assembly. For both chromosomes 10 and 11, capturing these genomic regions within single contigs indicates correction of the misassembled regions in the new assembly (Supplementary Figs. 3 and 4). The increased length and BUSCO completeness of the assembly, in addition to the

correction of putative misassembled regions, demonstrate the high quality of the assembly.

To annotate genes in the genome assembly, we combined ab initio gene prediction (24,583 genes and 27,697 isoforms, 98.8% BUSCO completeness) with a set of high-quality full-length transcripts obtained from isoform sequencing (35,187 transcripts, 94.5% BUSCO completeness). We merged both computational annotation and protein-coding transcripts from the full-length transcriptome into the final genome annotation that includes 25,167 annotated protein-coding genes with a total of 30,529 annotated isoforms. The new annotation featured the highest annotation completeness for amaranth to date (98.8% BUSCO completeness; Supplementary Table 1). The annotation of 23,155 isoforms was based directly on full-length transcripts from isoform sequencing and, therefore, includes high-confidence annotation of untranslated regions. A total of 219 Mb (50.43%) of the assembly was annotated as TEs and tandem repeats (Supplementary Table 2). Large parts of the genome were annotated as DNA transposons (98 Mb, 22.62%) and retroelements (109 Mb, 25.17%), including a high fraction of LTR elements (97 Mb, 22.33%; Supplementary Table 2). Corresponding to the previous report[32], LTRs and LINEs were mostly annotated in gene-sparse regions while DNA transposons were more evenly distributed along the genome (Supplementary Fig. 5).

We inferred the phylogenetic and syntenic relationships among species of the *Amaranthus* genus and with closely related species to study their genome evolution (Supplementary Fig. 6, Supplementary Fig. 7), which reflects previous results[33]. Peaks in the distribution of synonymous substitutions ($K_s$) indicate a single whole-genome duplication (WGD) in *Amaranthus* ($K_s \sim 0.5$) compared to *Beta vulgaris* ($K_s \sim 0.625$, Supplementary Fig. 6). We dated the WGD to 24.7 Mya (15.5–44.9 Mya), closely resembling the previous estimate of 25.56 Mya by Wang et al.[34]. Within the *Amaranthus* genus, genomes were highly co-linear with few large rearrangements (Supplementary Fig. 7).

### The interplay between chromatin state and methylation mediates TE silencing and gene expression

We used the improved reference genome to investigate the general structure of the chromatin landscape of amaranth by creating a genome-wide map of the chromatin landscape of *A. hypochondriacus* through ATAC sequencing (ATACome) and comparing it with other structures of the genome (Fig. 1A). We sequenced a total of eight samples from leaf ($n = 5$) and seedling ($n = 3$) tissue from the reference accession PI 558499 (Plainsman), yielding a total of 308,579,996 read pairs. From the 269,667,484 read pairs that uniquely mapped to the *A. hypochondriacus* reference genome (Supplementary Fig. 8), we identified a total of 142,649 accessible chromatin regions (ACRs) using the MACS3 pipeline[35]. Most ACRs were shorter than 1000 bp (Supplementary Fig. 9), with the largest at 5174 bp. We removed the two ACRs that were larger than 4000 bp for downstream analysis, as they might represent false positives and inflate ACR proportions. On average we identified 22,156 ACRs per sample, covering about 11,238,400 bp (2.56% of the reference genome, Supplementary Fig. 10). This fraction is similar to the range identified in other plant species and fits within the expected correlation between genome size and open chromatin[36,37] (Supplementary Fig. 11), suggesting that open chromatin can approximate functional space in the genome[38]. About 87.94% of ACRs physically overlapped between at least two samples (Supplementary Fig. 12). To reduce inherent variation in sequencing chromatin and create a more robust set of ACRs, we fused overlapping ACRs and discarded ACRs that were found only in one sample. This left the completed ATACome with 29,188 unique ACRs covering 23,071,669 bp or about 5.31% of the reference genome, representing the most complete overview of the functional space of amaranth and its genus to date.

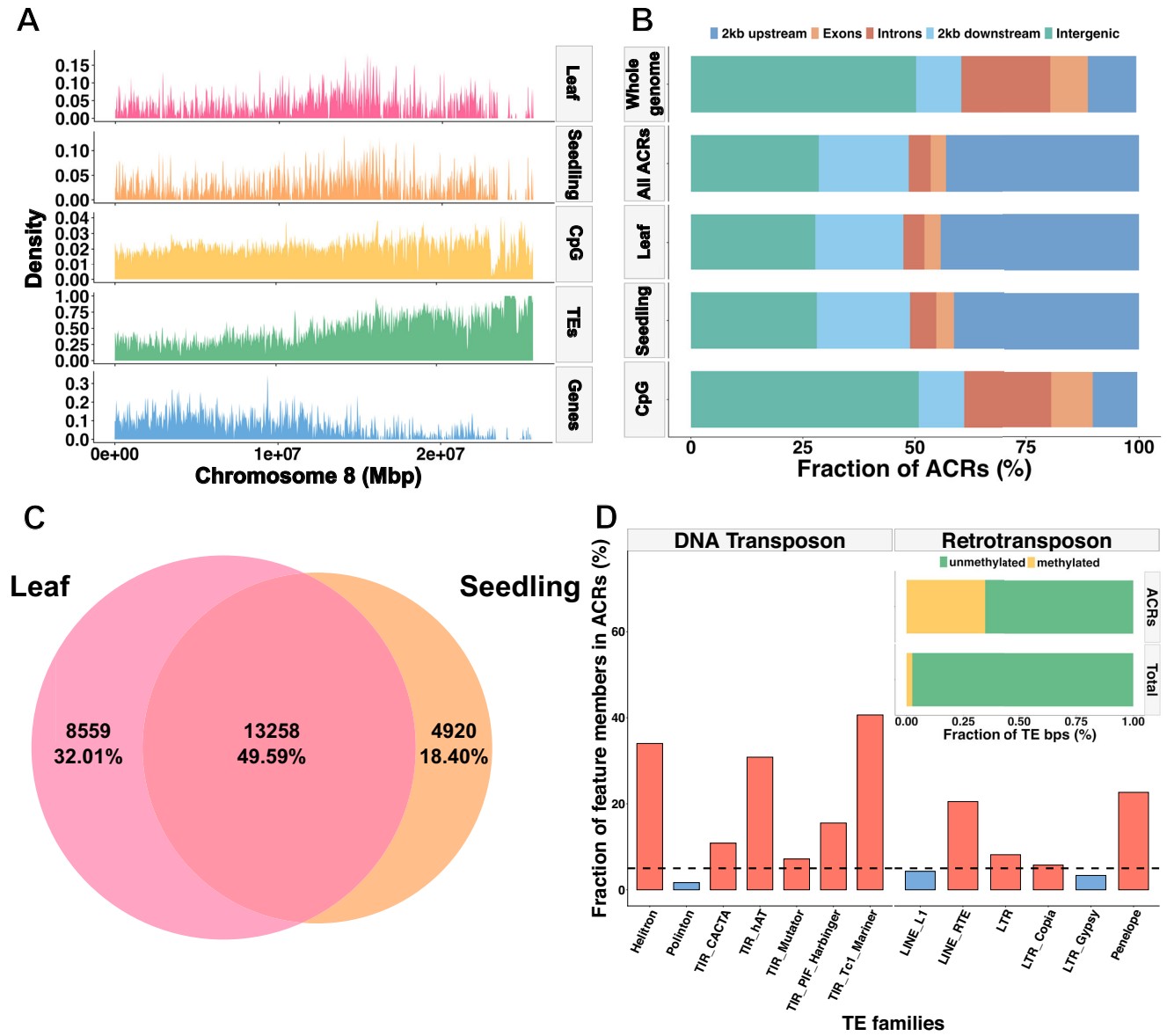

**Fig. 1 | Genome and chromatin landscape of *A. hypochondriacus*. A** Overview of the distribution of the gene content (blue), transposable element content (green), methylation content (yellow), accessible chromatin content in seedling tissue (orange), and accessible chromatin content in leaf tissue (pink) along chromosome 8. **B** Distribution of genome features in the whole genome, across all accessible chromatin regions (ACRs), leaf ACRs, seedling ACRs and methylation in CpG context. **C** Overlap of peaks called between seedling and leaf tissue samples. **D** Number of occurrences in ACRs for each of the 13 TE superfamilies and whether the superfamily is enriched (blue) or depleted (red) in open chromatin compared to the whole genome. Inset: Fraction of methylated (yellow) and unmethylated (green) base pairs for all TEs and TE base pairs within ACRs. Source data are provided as a Source Data file.

Investigating the landscape of these ACRs revealed a high association of ACRs with genes regardless of tissue (Fig. 1B). In total, 75.99% of ACRs were part of a gene body (intron: 4.32%, exon: 3.67%) or closely associated with it (2 kb upstream of transcription start site (TSS): 40.27%, 2 kb downstream of transcription termination site (TTS): 25.76%) (Fig. 1B). The distribution of ACRs among genome features was significantly different from a random distribution, likely reflecting the close association of ACRs with genes and their role in controlling gene expression[39]. Nearly half (49.59%) of the 30,304 ACRs were shared between the two tissues, documenting the within-plant variation in the chromatin landscape. Almost twice as many tissue-specific ACRs were identified in leaf tissue (8559) than in seedling tissue (4920), suggesting a higher differentiation of chromatin in leaves, compared to the seedling samples, which were comprised of multiple distinct tissues (cotyledon, hypocotyl, roots; Fig. 1C).

In addition to chromatin, methylation can control functional regions in the genome. As such, we generated a methylation map of *A. hypochondriacus* using methylation data obtained from whole-genome bisulfite sequencing (WGBS) of young leaf tissue from Plainsman using Bismark[40]. Of the 85,240,870 reads, 77,502,531 uniquely mapped to the reference genome. In total, 16.98% of Cs in the genome were methylated. Of these, 8.07% were in a CpG context, 4.63% were in a CHG context, and 4.28% were in a CHH context. Of the Cs in a CpG context, 75.77% were methylated, while 41.64% of Cs in the CHG context and 5.47% of Cs in the CHH context were methylated. Methylation in CpG, CHG and CHH contexts was higher in amaranth than observed in *Arabidopsis* but similar to grapevine, which has a similar genome size and TE content as amaranth[10,41,42]. About 50.82% of methylation accumulated in intergenic regions, 9.85% upstream of genes and 10.15% in gene bodies (Fig. 1B). Methylation in ACRs was

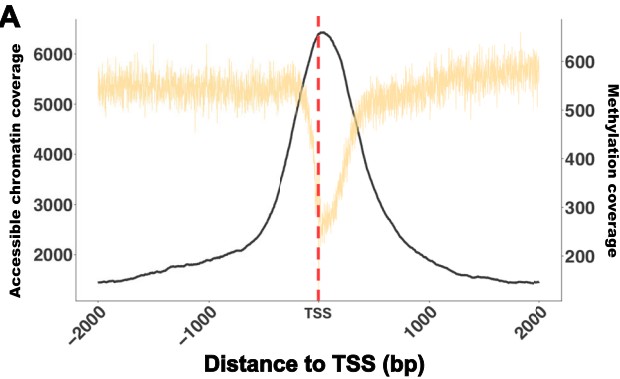

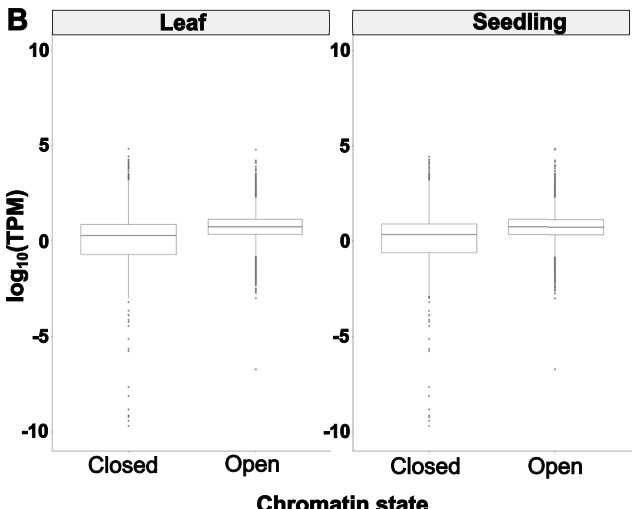

**Fig. 2 | Association between chromatin and gene expression. A** Accessible chromatin region (ACR) coverage (black) and methylation coverage (yellow) near the transcription start site (TSS). The red line indicates the TSS. **B** Comparison of mean expression (across 8 tissues) of genes associated with (within 2 kb of) accessible chromatin regions (open) and unassociated genes (closed). To avoid biases, an equal number ($n = 10,473$) of genes was randomly sampled without replacement for each combination of state and tissue. Letters indicate significant differences between groups based on a two-way ANOVA (tissue=0.011, state <2e-16, tissue:state=0.008) and followed by Tukey's test. Source data are provided as a Source Data file.

significantly depleted, with only 3.17% of open chromatin being methylated, indicating that the majority of accessible chromatin allows binding. Accordingly, the lowest methylation density was at the TSS, aligning with the highest density of accessible chromatin (21 bp downstream from the TSS) and probably facilitating the expression of genes (Fig. 2A).

Increased chromatin accessibility upstream of the TSS is often associated with higher expression of the corresponding gene. Indeed, we found that genes associated with tissue-specific ACRs showed significantly higher expression in their corresponding tissues compared to a similar sized set of randomly selected genes which were not associated with ACRs (Supplementary Fig. 13). This was the case for ACR genes in leaves and in seedlings (Fig. 2B). Despite the significantly higher expression of ACR genes, a gene ontology enrichment analysis did not find any significant terms that overlapped between tissues, but 26 enriched functions for seedling ACR genes and 13 enriched functions in leaf tissue ACR genes were identified (Supplementary Fig. 14). In leaf tissue, most enriched functions pertained to metabolic processes, while in seedling tissue 'signaling' and 'response to stimuli' were the most common.

The interaction of chromatin and TEs is also important for the functionality of plant genomes. TEs pose a threat to the integrity of the genome, due to their ability to insert into functional regions of the genome, potentially disrupting vital genes or other important regulatory elements[43]. Hence, the suppression of TE transposition by 'locking' them into condensed (closed) chromatin is likely one of the important functions of the chromatin state[44]. While most ACRs (78.83%) overlapped with TEs by at least 1 bp, the chromatin landscape was overall depleted for TEs compared to the rest of the genome. Indeed, TEs made up 47.74% of the *A. hypochondriacus* reference genome but only 35.23% of open chromatin (Supplementary Fig. 15). We found six DNA-transposon superfamilies and four retrotransposon superfamilies to be enriched in ACRs (Fig. 1D). This enrichment may result from the preference of these TE superfamilies to insert near gene bodies, which positions them within open chromatin and prevents their effective silencing through containment in closed chromatin[45]. Of the 63,782 TEs within open chromatin, 82.31% overlapped partially with ACRs, while 15.48% were completely open; another 2.21% of the TEs were large enough to carry entire ACRs with them. Therefore, the majority of ACR-associated TEs were not fully accessible and thus potentially inactive. Furthermore, methylation within ACRs was higher in TE sections, where 78.29% of methylated Cs were part of TEs. The number of methylated base pairs of TEs within ACRs was higher than for TEs outside of ACRs (Fig. 1D).

## Repeated enrichment of selective sweeps within differentially accessible chromatin during amaranth domestication

To elucidate the role that adaptation played in the divergence of the chromatin landscape between species, an accurate representation of the diversity within each species is needed. As such, we sequenced a total of 42 samples from leaf ($n = 22$) and seedling ($n = 20$) tissue from 18 accessions across five species, specifically *A. caudatus*, *A. cruentus*, *A. hypochondriacus*, *A. hybridus*, and *A. quitensis*, yielding a total of 1,710,942,952 read pairs (Supplementary Table 3).

Genome comparisons are often complicated by the need to align with a single reference that may be unevenly related to the studied individuals. To assess the potential of reference bias, we aligned our *A. hypochondriacus* ATAC-seq data to the *A. cruentus* reference genome[46]. We called a total of 156,378 ACRs on the *A. cruentus* genome, compared to 142,647 when mapping to *A. hypochondriacus*. By joining physically overlapping ACRs, we confirmed 30,792 unique ACRs covering 6.5% of the *A. cruentus* reference genome, similar to the 5.29% of the *A. hypochondriacus* reference. The association of *ACRs* was also seen in *A. cruentus*, where 68.63% of ACRs were found in (exon 5.13%, intron 5.6%) or near genes (2 kb upstream 38.51%, 2 kb downstream 19.39%); Supplementary Fig. 16). As in *A. hypochondriacus*, nearly half (46.37%) of the unique ACRs were found in both tissues, with 7,579 and 5,746 being unique to leaf and seedling tissue, respectively (Supplementary Fig. 17). Independent *t*-tests comparing ACRs called from the two reference genomes showed that neither the number of ACRs called ($p = 0.4138$), the distribution of ACRs along the genome ($p = 1$), nor the distribution among tissues ($p = 0.8822$) significantly differed between the different reference genomes. Together, these results suggest no significant impact of reference bias when calling ACRs of different grain amaranth species. Hence, we carried out our multiple species comparisons using the new *A. hypochondriacus* genome.

Across all samples of the five species, 79% of the reads uniquely mapped to the *A. hypochondriacus* reference genome. A principal component analysis (PCA) of single nucleotide polymorphisms called from ATAC-seq data reconstructs the population structure similar to the whole genome sequencing data (Supplementary Fig. 18B). Calling coverage peaks to identify accessible chromatin from this data identified a total of 178,194 ACRs, covering an average of 2.38 to 2.67% of the reference genome (Fig. 3A). A PCA based on ACR presence-absence separated the samples stronger by tissue than by species, contrary to

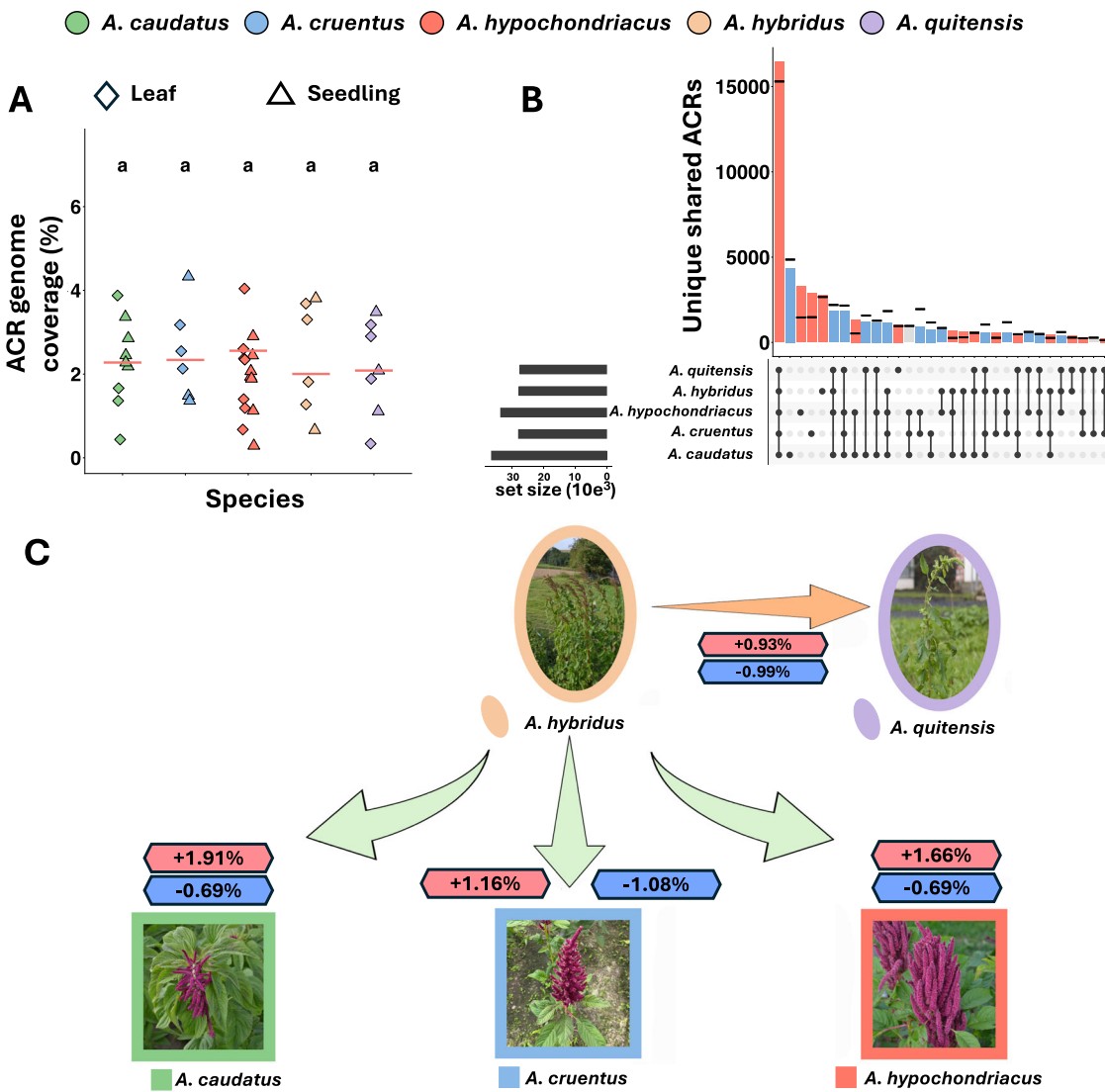

**Fig. 3 | Chromatin changes during amaranth domestication. A** Fraction of genome covered by ACRs in each sample. Red lines indicate the mean coverage per species (*A. caudatus* n = 9, *A. cruentus* n = 6, *A. hypochondriacus* n = 14, *A. hybridus* n = 6, *A. quitensis* n = 7). Letters above the species indicate significant differences between groups based on a one-way ANOVA (species=0.821) followed by Tukey's test. **B** Number of unique and shared ACRs across species. Colors indicate whether the groups are enriched (red), depleted (blue) in ACRs or within (gray) the confidence intervals (black lines). **C** Fraction of genome in differentially accessible chromatin between wild *A. hybridus* and grain amaranth species that opened up (red boxes) or closed (blue boxes) compared to the chromatin landscape of *A. hybridus*. Source data are provided as a Source Data file.

the PCA on genotypic data, but aligning with the important developmental function of chromatin (Supplementary Fig. 18A). An ANOVA revealed that there was no significant difference between species in total amount of open chromatin (Fig. 3A) or total number of ACRs (Supplementary Fig. 19). The merging of disparate ACRs from all species resulted in 51,571 distinct ACRs covering a total of 10.2% of the reference genome. ACRs that were shared by all species were the most frequent (31.07%, Fig. 3B), similar to the large overlap between tissues (Fig. 1C). The next most common groups were ACRs unique to each of the three domesticates (5.61–10.32%) and unique to their wild ancestor (5.05%). This indicates that most changes within the chromatin landscape that occurred during domestication were species-specific (a total of 32.43% of all ACRs), reflecting the separate domestication processes. In addition, nearly 10% of ACRs were shared between two or more domesticates, but not with a wild relative, indicating that a large fraction of the ACRs changed during domestication. To control the observed overlaps of ACRs between species for unequal sample size and structure, we permuted species assignments for every species combination (Fig. 3B). For 29 of the 31 species combinations, the ACR

count was outside the 95% confidence interval, suggesting that they are not randomly overlapping. Comparing each of the domesticated species (*A. caudatus*, *A. cruentus*, and *A. hypochondriacus*) to the wild ancestor *A. hybridus*, we identified between 13,856 and 15,742 total ACRs that changed their state (closed-open and open-closed, respectively). These differentially accessible chromatin regions (dACRs) that were potentially altered during the domestication process made up between 2.24% to 2.6% of the reference genome (Fig. 3C). In all three of the domesticated species, more chromatin opened (*A. caudatus*: 1.91%, *A. cruentus*: 1.16%, *A. hypochondriacus*: 1.66%) than closed (*A. caudatus*: 0.69%, *A. cruentus*: 1.08%, *A. hypochondriacus*: 0.69%) during the speciation of the domesticates. In comparison, between the two wild relatives (*A. hybridus* and *A. quitensis*) 0.93% opened and 0.99% closed, differing from the pattern observed during domestication (Fig. 3C). Together, this suggests that during the domestication more chromatin opened than closed, but not between the two wild species.

The significant number of dACRs identified suggests that specific chromatin regions might have been under selection during the domestication process (Fig. 4). To test for signals of selection, we

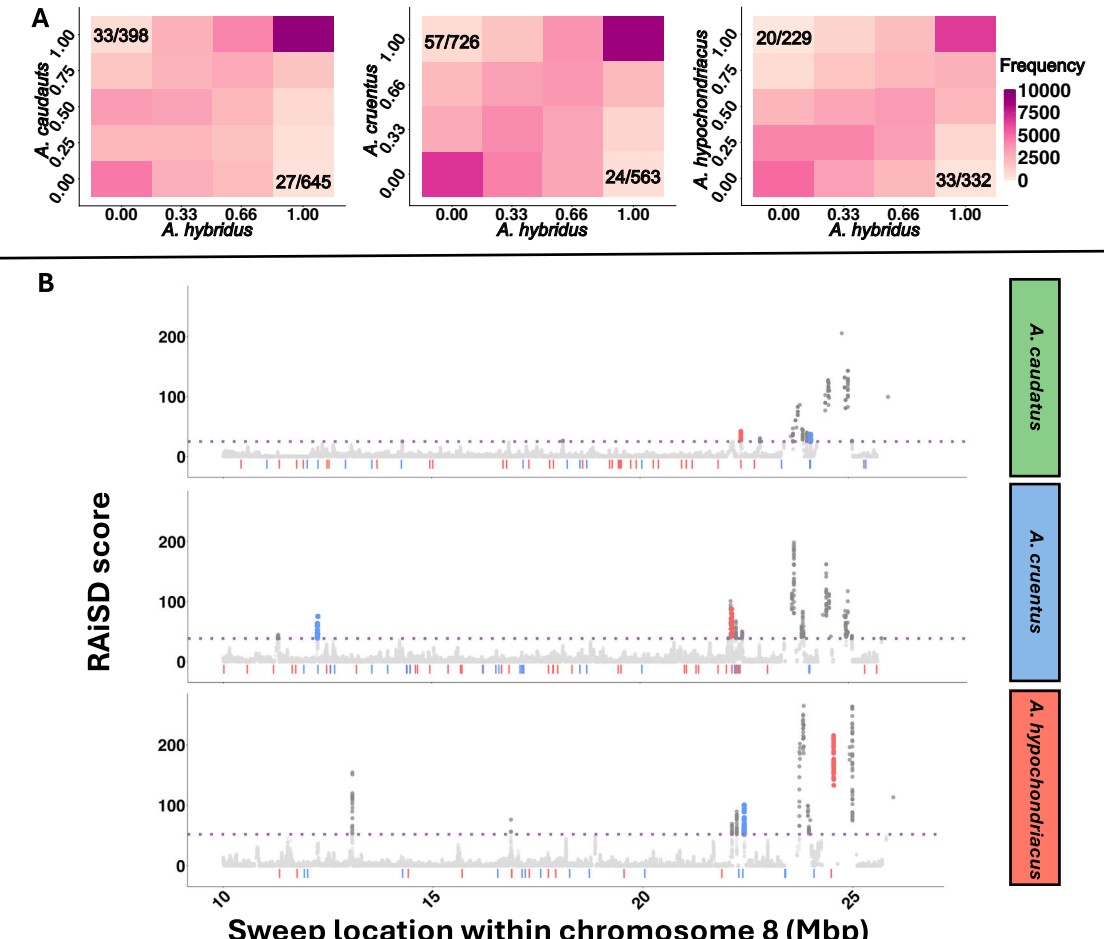

**Fig. 4 | Selection on chromatin during domestication. A** Joint frequency spectrum of ACRs in domesticates and *A. hybridus*. x and y axis indicate the ACR frequency across accessions in the respective species. **B** Overlap of fixed dACRs that opened during domestication (red) or closed (blue) and selective sweeps in a subsection of chromosome 8. The purple dotted line indicates the significance threshold for selective sweeps. Source data are provided as a Source Data file.

analyzed ACR frequency changes between the wild *A. hybridus* and each of the crop species. In total, we found 1600 dACRs that were fixed closed in the wild ancestor (*A. hybridus*) and fixed opened in the crop species (*A. caudatus*: 645; *A. cruentus*: 726; *A. hypochondriacus*: 229) and 1293 dACRs that were closed and fixed during domestication (*A. caudatus*: 398; *A. cruentus*: 563; *A. hypochondriacus*: 332; Fig. 4A). There was no significant overlap between fixed differences (wild-domesticated) across crops species (Supplementary Fig. 20), indicating that selected dACRs were species specific. This agrees with the previous finding based on SNP-based selective sweeps, which also differed among the three domesticated species[27].

To further elucidate the selective role of dACR regions during domestication, we investigated their association with selective sweeps. We found that between 3.41 and 6.63% of domestication dACRs overlapped with selective sweeps in the respective species (Fig. 4A). Of the dACRs within selective sweeps, 46 were associated with genes (within 2 kb from TSS and TTS). Of these, we found six that opened and nine that closed in *A. caudatus*, six that opened and 11 that closed in *A. cruentus*, and seven that opened and seven that closed in *A. hypochondriacus* (Supplementary Table 4). A hyper-geometric test showed a significant enrichment of selective sweeps within dACRs in all three domesticates. While none of the genes that opened during domestication were shared between the species, one of the genes that closed was shared between *A. caudatus* and *A. hypochondriacus*, two were shared between *A. cruentus* and *A. hypochondriacus* (including a disease resistance gene *AHq015015*), and three were shared among all three domesticates. One of the genes that closed in all three

domesticates was also a disease resistance gene (*AHq015012*). Two of them belonged to the HAUS augmin-like complex (*AHq020586* and *AHq020587*) coding for the subunit 1 (AUG1), which has an essential developmental function. The potential involvement of the HAUS augmin-like complex in crop domestication and the potential influence of chromatin state on its function should be further investigated through molecular assays.

To investigate the functional link between chromatin accessibility and phenotypic changes, we looked at seedling coloration in *A. caudatus*, a trait with mendelian inheritance[30]. The red pigmentation through betalains is controlled by the expression of the key enzyme CYP76AD2[31]. We compared the chromatin state between red seedlings and green seedlings and found that upstream of the gene, in the potential promoter region, red seedlings had a stronger accessible chromatin signal than green seedlings (Supplementary Fig. 21). This connection between dACRs and phenotypic differences illustrates the probable importance of chromatin changes for trait changes. Through dissecting the genetic control of domestication traits in amaranth more links between dACRs and phenotypes can be revealed.

## Discussion

We present a population-scale investigation of chromatin-landscape changes during amaranth domestication. Our study provides the most complete genome assembly, methylome and chromatin map of grain amaranth (*A. hypochondriacus*), enabling functional studies in the orphan-crop and the *Amaranthus* genus as a whole that were previously hindered by the lack of annotation of functional regions and

gene models[30,31,47]. The quality of our assembly and chromatin map is comparable to that of major crops[23,36,37].

Our results highlight how chromatin is integrated into the broader genomic landscape and contributes to the genome ecosystem[48]. We demonstrate that approximately two-thirds of ACRs are closely associated with genes across tissues, with approximately 40% of ACRs being associated with the promoter region alone. The high density of ACRs upstream of genes emphasizes the functional significance of promoter regions and illustrates the utility of chromatin profiling in delineating functional genomic elements beyond coding sequences[49,50]. Nonetheless, ACRs distant to genes might also be part of the functional space, as distal ACRs often contain enhancers, sRNAs and other regulatory regions[50,51]. Accessible chromatin enables transcription factor binding and gene expression[37], but also increases the proliferation of TEs[45]. While we find a large overlap between TEs and ACRs, most TEs were only partially open (less than 25% of TE in ACRs). Of the TEs that were located in ACRs, a large fraction (32.4%, compared to 4.49% genome-wide) was methylated, which might lead to their silencing even when they are located in an ACR. Whether the over-representation of certain TE superfamilies in open chromatin regions results from their preferential insertion into gene-rich areas or is facilitated by the accessibility of the chromatin in those regions remains unclear[48]. Further characterization of the genomic landscape will be critical to understand the interplay between genetic elements and other genomic properties, including methylation, chromatin state, and recombination rates.

Characterizing the chromatin landscape at the population level provides insights into its role in evolutionary processes[50,52]. During the domestication of the grain amaranths, the total amount of accessible chromatin remained roughly the same across the five species studied; however, ~2.5% of chromatin changed state in each of the three domestication processes. In all three transitions, a higher fraction of the chromatin opened, indicating a trend towards more open chromatin (Fig. 3C), consistent with findings in soybean[23], maize[53], and rice[9]. Genes associated with ACRs are expected to exhibit higher expression levels compared to non-associated genes[20](Fig. 2B). Considering the increased level of open chromatin accessibility, an overall increase in gene expression is expected during domestication. Similar patterns have been seen in other domesticated species, such as maize[54] and cotton[55], in which an increase in overall gene expression has been observed. This effect is particularly pronounced for genes governing traits that were directly under selection[56]. These regulatory modifications are reflected in the histone modification and DNA conformation[23].

The impact of chromatin accessibility changes on phenotypes could drive their selection, either through genomic mutations that lead to altered chromatin or through epigenetic remodeling. Our finding of a significant overlap between dACRs and selective sweeps indicates that chromatin did indeed play a role during domestication. Further research is needed to determine whether selection acts on chromatin state to cause a causal trait change or if the chromatin change is simply a side effect of mutations, without directly altering the traits. The group of resistance genes and developmental regulators that we found might be promising candidates for investigating the complex interactions among genomic mutations, chromatin dynamics, trait expression, and selection processes during domestication.

Directional selection during the domestication of plants and animals has long served as a model to understand evolutionary changes[57]. Despite the clear impacts of domestication on phenotypes, intrinsic traits, such as the chromatin landscape, seem to have responded to selection[58–60]. This response was either through the direct control of domestication traits by the chromatin state or through a pleiotropic relationship between phenotypes and chromatin state. Assessing the diversity of chromatin and its functional importance could unlock an additional set of variation for crop improvement.

## Methods

### Genome assembly and annotation

High molecular weight (HMW) DNA was extracted from fresh leaf tissue of *A. hypochondriacus* (PI 558499; cv. Plainsman) using a CTAB-Genomic-tip protocol[61]. The HMW DNA was quantified with the Nanodrop™ One/OneC Microvolume UV-Vis Spectrophotometer (Waltham, MA), and screened for quality control parameters including DNA concentration (>800 ng/mL) and contamination (260/280 and 260/230 ˜ 2.0). For PacBio HiFi sequencing, HMW DNA was sheared to 17 kb on a Diagenode Megaruptor (Denville, NJ) and then made into SMRTbell adapted libraries using a SMRTbell Express Template Prep Kit 2.0 (Pacific BioSciences, Menlo Park, CA), followed by size selection using a Sage Science BluePippin (Beverly, MA) for fragments greater than 10 kb. Sequencing was done at the Brigham Young University DNA Sequencing Center (Provo, UT, USA) using Sequel II Sequencing Kit 2.0 with Sequencing Primer v5 (PN:102-067-400; PacBio; United Kingdom, London, UK) and Sequel Binding kit 2.2 for 30 h with adaptive loading using PacBio SMRT Link recommendations. For Oxford Nanopore Technology (ONT) sequencing, HMW DNA was sequenced using the default protocol for ultra-long DNA sequencing kit (SQK-ULK114; Oxford Nanopore Technologies, Oxford, UK) following the standard manufacturer's protocol with a starting volume of 750 μL (250 ng/μL) on a R10.4.1 flow cell (FLO-MIN114; ONT, Oxford, UK) for 24 h. Flow cells were washed and reloaded three times. Dorado v0.5.0 + 0d932c0 with the model 'dna_r10.4.1_e8.2_400bps_sup@v4.2.0' was used for base calling of POD5 files. Chopper v0.5.0[62] was used for trimming (<30 kb) and quality control (qv <8.0). In total, approximately 30X coverage of both HiFi (N50 = 14 kb) and ONT reads (N50 = 49 kb) was generated and used for the primary contig assembly.

A primary contig assembly was generated using hifiasm v0.19.8[63] with default parameters for an inbred species (-l0). Scaffolding of the primary assembly into pseudo-molecules was accomplished using publicly available Hi-C (high-throughput chromosome conformation capture) data (SRA accession: SRR5345531[30]). Hi-C reads were aligned to the primary contig assembly using the Burrows-Wheeler Aligner[64] following the Arima-HiC mapping pipeline (A160156_v03, Arima Genomics, Carlsbad, California, USA). Only uniquely aligned paired-end reads were retained for downstream analyses. Contigs were then clustered, ordered, and oriented into scaffolds using YaHS[65] followed by manual inspection and correction using JuiceBox[66]. The corrected scaffolded assembly was evaluated and corrected with Inspector[67] to produce the final assembly. We named the assembled pseudochromosomes according to scaffold names of *A. hypochondriacus* reference genome v2 and ensured colinearity with the previous assembly version[30,31]. To prepare the genome for genome annotation, we generated a repetitive element database using RepeatModeler[68] and softmasked the assembly using Repeatmasker[69].

To annotate the *A. hypochondriacus* assembly v3, we combined an ab initio gene prediction generated using BRAKER3 v3.0.8[70] guided by protein and RNA-seq evidence with full-length PacBio Iso-Seq transcripts. As a protein database, we used the protein sequences of 117 embryophyta species from ODB10[71] in addition to annotated proteins of *A. cruentus*[46]. As RNA-seq evidence, we used a public dataset of 31.7 Gb of 90 bp paired-end HiSeq Illumina sequencing data from eight different tissues (BioProject accession PRJNA263128)[72]. To aid the alignment of RNA-seq reads, we generated a preliminary BRAKER3 gene prediction guided only by protein evidence. We mapped the RNA-seq reads to the assembly with STAR v2.7.8a[73] using the preliminary gene prediction as the splice-junction database. We then generated a second BRAKER3 gene prediction guided by both protein and RNA-seq evidence.

For the assembly of full-length Iso-seq transcripts, we used two datasets, from multiple tissues of accession PI 558499: We used a total of 5.3 million circular consensus sequencing (CCS) reads (mean length:

1441 bp) of public data from seven tissues, root, cotyledon, flower, leaf, pollen, developing seed, and mature seed (BioProject ID PRJEB65083)[31]. We further generated 4.4 million CCS reads (mean length: 1,711 bp) from root, leaf, stem, inflorescence and whole seedling tissues of the *A. hypochondriacus* reference accession Plainsman (PI 558499). RNA samples were extracted from each tissue type independently using the Zymo Research (Tustin, CA, USA) Direct-zol RNA MiniPrep Plus kit (R2072). The quantity and quality of extracted RNA were evaluated for quality using a Bioanalyzer 2100 (Agilent Technologies, Santa Clara, California, USA). After quality check, RNA from each of the different tissues was pooled in equal molar ratios to synthesize full-length complementary DNA (cDNA) using a NEBNext® single cell/low input cDNA synthesis and amplification kit (E6421L), which uses a template switching method to generate full-length cDNAs (New England BioLabs, Ipswich, MA, USA). IsoSeq libraries were prepared from the cDNA of *A. hypochondriacus* (PI 558499; cv. Plainsman) according to standard protocols using the SMRTbell v3.0 library prep kit (Menlo Park, CA, USA) and sequenced on a single SMRT cell 8 M for each species using a PacBio Sequel II at the DNA sequencing center at Brigham Young University (Provo, Utah, USA).

For both datasets, we processed Iso-Seq CCS reads using Isoseq3 (https://github.com/PacificBiosciences/IsoSeq) and clustered full-length non-chimeric (FLNC) reads into transcripts using Isoseq3 *cluster*. To further deduplicate both datasets, we mapped clustered transcripts to the genome assembly using minimap2 v2.26[74] with the *splice:hq* preset and collapsed them into sets of unique transcripts using cDNAcupcake v28.0.0 (https://github.com/Magdoll/cDNA_Cupcake) with minimum cutoffs for coverage 0.95 and identity 0.9, resulting in 53,150 and 57,731 transcripts from the public and newly generated datasets, respectively. We used the *chain_samples.py* script of cDNAcupcake to combine both datasets, allowing for a maximum 3' end difference of 300 bp and a fuzzy splice junction distance of 5 bp. We used SQANTI3 v5.2[75] to correct the combined transcript sequences based on the sequence of the genome assembly. To filter potential artifacts from the combined transcript set, we required transcripts not found in the BRAKER3 prediction to be supported by at least five CCS reads and at least 5% of CCS reads per locus, resulting in 35,187 transcripts (median length: 1802 bp, 94.5% BUSCO completeness). To combine full-length transcript sequencing data and BRAKER3 prediction, we predicted open reading frames in the filtered transcript set using GeneMarkS-T[76] and merged both datasets using TSEBRA v1.1.2.4[77]. We assessed completeness of the Iso-Seq transcripts and the final annotation using BUSCO v5.2.2[78] in protein mode against the Embryophyta reference set of OrthoDB v10[71]. For functional annotation of annotated genes, we submitted the annotated protein sequences to Mercator4 v6.0[79] and eggNOG-mapper v.2.1.12[80]. To annotate repetitive elements in the genome assembly, we ran the annotation pipeline EDTA v2.2.1[81] with the 'sensitive' parameter and included annotated CDS for purging of gene sequences from the TE library with the *cds* parameter.

## Phylogeny, whole genome duplication and synteny analysis

An orthogroup analysis was constructed with Orthofinder2 v.2.5.4[82] from longest protein-coding gene models for six *Amaranthus* species with fully assembled genomes (*A. tricolor*[34], *A. tuberculatus*[83], *A. palmeri*[83], *A. retroflexus*[83], *A. cruentus*[46] and *A. hypochondriacus*) and two outgroup species within the family Amaranthaceae, *Beta vulgaris* L.[84] and *Chenopodium quinoa* Willd[85]. Genomes for *A. tuberculatus*, *A. palmeri* and *A. retroflexus* were acquired from the WeedPedia Database[86]. Orthofinder2 assigned 95.1% of all genes to an orthogroup, with a G50 and O50 of nine and 6,919, respectively. A rooted species tree phylogeny was produced using the multiple sequence alignment approach of OrthoFinder2, elicited with the "-M msa" option to produce bootstrap values. WGD v.2.0.38[87] was employed to identify WGDs and speciation events by leveraging synteny inference and heuristic

peak detection with 95% confidence intervals derived from anchor gene pairs. Synonymous substitution rates (Ks) were calculated between paralogous gene pairs to infer WGDs and between orthologous gene pairs to infer speciation events within and among species. Divergence times among species and WGD events were inferred using a synonymous substitution rate for Amaranthaceae of 9.6E-9[34]. Wang et al. reported the divergence between *A. tricolor* and *B. vulgaris* at a Ks peak of approximately 0.63 (divergence between Amaranthoideae and Chenopodioideae subfamilies) and that *A. tricolor* diverged from *B. vulgaris* approximately 32.81 Mya. That translates into an estimated substitution rate of 9.6E-9, which was calculated using the following equation[34]:

$$\text{Divergence time (Mya)} = [Ks/(2 * \text{Substitution rate})] * 10\text{-}6 \quad (1)$$

Syntenic relationships across the *Amaranthus* species were visualized in SynVisio (https://synvisio.github.io/#/) from orthologous genes in collinear blocks, identified with blastp and McScanX[88] using the parameters "-e 1e-50 -k 25 -s 50".

## Methylation sequencing

Whole genome bisulfite sequencing (WGBS) was generated from young leaf tissue from a single plant from the *A. hypochondriacus* accession Plainsman (PI 558499). Tissue was collected, freeze-dried, and DNA extracted using a modified mini-salts extraction protocol[89]. Quality control parameters for concentration (>300 ng/mL) and contamination (260/280 and 260/23 ~ = 2.0) were followed before sequencing. The DNA samples were sent to Novogene Corporation, Inc. (San Diego, CA) for WGBS. In brief, the genomic DNA (spiked with lambda DNA) was fragmented to 200-400 bp and then subjected to bisulfite to generate single-strand DNA using a EZ DNA Methylation Gold Kit (Zymo Research, Irvine, CA). During the bisulfite treatment, unmethylated cytosine is converted into uracil, while methylated cytosine remains unchanged. Methylation sequencing adapters were ligated, followed by double strand DNA synthesis using the AccelNGS Methyl-Seq DNA Library kit (Swift Biosciences, Ann Arbor, MI). The quality of the library was verified with Qubit and real-time PCR and the size distribution was verified on a bioanalyzer. Libraries were pooled and sequenced on Illumina NovaSeq X Plus (PE150) instrument to produce a minimum of 30X coverage according to genome size.

We aligned the methylation data to the new reference genome by running Bismark[40] with following parameters "– bowtie2 -n 0 -l 20 methylation/". The reference genome methylation index was prepared using the bismark_genome_preparation function. The final summary of aligned ACRs was performed using the Bismarck's methylation extractor function. Lastly, we calculated methylation density 2000 bp up- and downstream from the TSS using an in-house script.

## Chromatin accessibility

In order to assess chromatin accessibility changes during domestication, we performed ATACome of a total of 42 samples. These consisted of 3 accession of *A. cruentus* L., 4 accessions of *A. caudatus* L., 4 accessions of *A. hypochondriacus* L., 3 accession of the wild ancestor *A. hybridus* L., and 4 accessions of the wild (potentially feral) *A. quitensis* Kunth. (Supplementary Table 3). We sampled leaf tissue from plants grown for 32 days under short-day conditions (8 h light, 16 h dark) at 22 °C and seedling tissue from plants grown on filter paper in petri dishes in the dark for 7 days for nuclei extraction.

We extracted nuclei from approximately 50 mg of leaf tissue from the fourth fully developed leaf and 100 mg of whole seedlings[90]. In brief, tissue samples were placed in 0.5 mL of chilled lysis buffer (15 mM Tris-HCl pH 7.5, 20 mM NaCl, 80 mM KCl, 5 mM Dithiothreitol (DTT), 0.5 mM Spermine, 1x Cocktail (ThermoFischer: 78429), 0.2% Triton X-100) and finely chopped with a razor blade to release nuclei. The nuclei suspensions were stained with 2 μL of 1 mg/mL 4,6-Diamidino-2phenylindole (DAPI). After confirming the presence of intact nuclei under a fluorescence microscope at 20x magnification, we

selected 50,000 intact nuclei based on their size and intensity of DAPI signal under 488 nm excitation using a FACSVantage SE (Beckon Dickinson) and collected them in 0.5 mL lysis buffer. We prepared the 50,000 nuclei for tagmentation by centrifuging them for 4 min at 1000 g and 4 °C and discarding the supernatant. Afterwards, we washed the pellet in 1 mL of wash buffer (10 mM Tris-HCl pH 8.0, 5 mM MgCl2, 1x Cocktail) and centrifuged for 4 min at 1000 g and 4 °C and discarded the supernatant. We then mixed the nuclei with 25 μL of Tagment DNA Buffer and 2.5 μL TN5 (Illumina: 20034917) and incubated them at 37 °C for 30 min. The tagmented samples were purified using the QIAGEN MinElute kit (QIAGEN: 28004). The prepared libraries were sequenced for a minimum of 10 million 100 bp paired-end reads per library on a NovaSeq 6000 platform by the Cologne Center for Genomics (CCG).

## ATAC-seq data processing

We aligned the sequencing reads to our new *A. hypochondriacus* reference genome using default parameters of bwa-mem2 (2.2.1)[91]. Duplicate reads were identified using Picard (2.27.5) and removed with the REMOVE_DUPLICATES = TRUE setting[92]. To exclude over-represented regions in the genome, we employed a read depth cutoff of 250. ACRs were called using the callpeak function of macs3 (v3.0.0)[35] in BAMPE mode with a reference genome size of 439 Mb and a false discovery rate of 0.01. We excluded peaks that only occurred in one library per species and that were longer than 4000 bp from further analyses as potential false positive calls. To normalize the peak calling for mapability, we also called peaks for whole genome sequencing (WGS) data of each of the 18 accessions[27] and removed ACR peaks from our ATAC-seq data that were also called in the whole genome sequencing data.

## ATACome for PI 558499

We assembled a high-confidence ATACome for *A. hypochondriacus* from 8 samples of the reference accession PI 558499 consisting of five leaf and three seedling tissue samples. In addition to the filters above, we only considered ACRs, which could be called in at least two of the eight samples, to reduce false positives.

We overlapped the ACRs genome-wide with other genomic features, i.e., gene density, TEs and methylation signatures using the genomicDensity function from the circlize package[93]. Genomic feature annotation of ACRs was performed using the assign-ChromosomeRegion function of the ChIPpeakAnno package in R[94]. Analysis of the distribution of genomic features for the whole reference genome using assignChromosomeRegion was performed using non-overlapping 10 bp windows due to vector size constraints in R studio. Boundaries were set to 2000 bp for the upstream (upstream = 2000, downstream = 0) of TSS and downstream (upstream = 0, downstream = 2000 of TES) genomic features of assignChromosomeRegions. We discarded ACRs that were called in less than two samples of the same tissue to study tissue-specific ACRs. We investigated overlaps between tissues with the findO-verlapsOfPeaks function of the ChIPpeakAnno package[94] and identified tissue-specific ACRs. ACR density 2,000 bp up and downstream of the TSS was calculated using a custom script (https://git.nfdi4plants.org/stetter-lab/chromatin_amaranth_2025). To investigate a potential enrichment of gene functions in open genes, we performed a GO-enrichment analysis, employing the goseq package in R[95]. Peak-length distribution for both tissues was calculated using a custom script (https://git.nfdi4plants.org/stetter-lab/chromatin_amaranth_2025). We compared gene expression between genes associated with open (ACR within 2000 bp of TSS) and closed chromatin within and between tissues, by quantifying gene expression in publicly available data from eight different tissues[72] using kallisto v0.48.0[96], calculating the mean expression for each gene across the eight tissues and performing an ANOVA to test for significant differences and corrected for

multiple testing using Tukey's HSD. The expression values were transformed using log10(TPM + 0.001). Only genes that were associated with ACRs unique to leaf or seedling tissue were considered. To prevent biases by different-sized datasets, the smallest set of genes with expression data was identified among the two tissues and chromatin states (open seedling with 10,473 ACR-associated genes), and the same number of genes was randomly sampled from the other three sets. The same conditions were used to compare expression data between chromatin state in each of the eight tissues from Clouse et al.[72], with the smallest set of genes with expression data being cotelydons (10,793 genes). We then determined all genes with ACRs within the gene body or their promoter region (2000 bp upstream), using the annotatePeakInBatch function of ChIPpeakAnno (output = overlapping, maxgap = 2000)[94]. The gene expression of ACR-associated genes (open genes) was then compared to the expression of an equal number of random genes without an associated ACR (closed genes) using an ANOVA. To study the association between TEs and accessible chromatin, we overlapped ACRs with our TE annotation using the annotatePeakInBatch function from the ChIPpeakAnno package[94]. Enrichment and depletion of TE superfamilies in accessible chromatin was determined by performing hypergeometric tests in R using the phyper function from the stats package[97]. Lastly, we calculated the fraction of methylation within ACRs, TEs and TEs within ACRs. Enrichment of methylated base pairs within each of these three groups was tested by performing a hypergeometric test.

## Testing for reference bias

To explore whether aligning ATAC-seq data of multiple species to the *A. hypochondriacus* reference genome causes a bias in peak calling towards data from *A. hypochondriacus* accessions, we also aligned all ATAC-seq data to the *A. cruentus* reference genome[46]. These alignments were subject to the same data processing parameters as for *A. hypochondriacus* aligned data, except for adjusting the genome size during peak calling to the 370.9 Mb of the *A. cruentus* assembly[46]. A chi-square test was performed to test whether the ratio of called peaks between the species is the same between the two alignments. Additionally, chi-square tests were performed to identify if the peak distribution between the two tissues or between genomic regions significantly differed between the two alignments.

## Species comparison of open chromatin

To compare ACRs between species, we used only ACRs that occurred in at least two samples of a species, regardless of tissue. Overlapping ACRs from different samples within a species were joined with the reduce function of the GenomicRanges package[98], to create five deduplicated lists of all ACRs within each species. To uniquely index ACRs across species, we combined the species-specific lists into one with unique ACRs for the *Amaranthus* family, by joining ACRs that overlapped between species with the reduce function of the GenomicRanges package. These IDs were then assigned to corresponding ACRs within each species-set. We determined ACR changes in pairwise comparisons between species, based on IDs. To test for statistical significance, we randomized the species identity of each ACR for each sample using the sample function in R and inferred the distribution of ACRs. We calculated confidence-intervals from 100 permutations, using the t.test function of the stats package[99] in R. Differential accessible chromatin regions (dACRs) for each of the three domesticated species compared to the wild *A. hybridus* were identified through a custom script (https://git.nfdi4plants.org/stetter-lab/chromatin_amaranth_2025) and enrichment of gene functions in dACRs was tested using the goseq package.

## Overlap between selection signals and accessible chromatin

We called selective sweeps for each of the five species based on whole genome sequencing data of 28 *A. caudatus*, 21 *A. cruentus*, 18 *A.*

*hypochondriacus*, 9 *A. hybridus* and 12 *A. quitensis* accessions, respectively[100]. We used RAiSD[101] to identify potential selective sweeps with a cut-off at 0.01% of the highest RAiSD scores. The overlap between selective sweeps and dACRs was determined using bedtools intersect[102]. The enrichment analysis on the selective sweep associated with dACRs was performed using goseq. An enrichment of selective sweeps in dACRs was confirmed by testing the overlap of bp in selective sweep regions within ACRs compared to the whole genome, using a hypergemetric test[97].

**Chromatin accessibility near key pigmentation gene**
We calculated the mean depth for 10 kb up and downstream of CYP76AD2 (chromosome 16: 5238629-5258629 bp) from our ATAC-seq seedling samples of three *A. caudatus* accession PI 490518, PI 608019, PI 642741 with red seedling color and PI 490612 with green seedling color, respectively. The read count was normalized by dividing the depth by the average depth over the 10 kb. The normalized depth was then overlapped with the 41,812 ACRs found within *A. caudatus*. We plotted the mean normalized depth along the region to examine regions that are differentially open.

**Reporting summary**
Further information on research design is available in the Nature Portfolio Reporting Summary linked to this article.

## Data availability
All generated sequencing data is available through the European Nucleotide Archive (ENA) under the Bioproject PRJEB88670. The newly assembled *A. hypochondriacus* genome and annotation files are accessible at AmaranthGDB [https://amaranthgdb.org/downloads.html]. Previously reported RNA-seq data[72] is available under Bioproject PRJNA263128. Circular consensus sequencing (CCS) data can be accessed under BioProject PRJEB65083. Hi-C data can be accessed under the SRA accession SRR5345531 Source data are provided with this paper.

## Code availability
Scripts used in the analyses are available through Github [https://github.com/cropevolution/chromatin_landscape_amaranth_2025] and DataPLANT [https://git.nfdi4plants.org/stetter-lab/chromatin_amaranth_2025].

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

## Acknowledgements

We thank Roswitha Lentz and Christoph Göttlinger for their help with ATAC-library preparation. We thank Jeffrey Ross-Ibarra for valuable feedback on the manuscript. We acknowledge funding by the Deutsche Forschungsgemeinschaft (DFG, German Research Foundation) under Germany´s Excellence Strategy – EXC-2048/1 – Project ID 390686111 and grant STE 2654/5 to MGS by the DFG.

## Author contributions

M.G.S. conceptualized and designed the study. C.G. collected and analyzed the chromatin data. P.J.M. generated data and assembled the reference genome and performed the synteny analysis. P.J.M. collected methylation data. T.S.W. performed annotation of the new reference genome. C.G., T.S.W., and P.J.M. prepared the figures. All authors wrote, edited and approved the manuscript.

## Funding

## Competing interests

The authors declare no competing interest.
