## [Peer Review file · Nature Communications]

Domestication shaped the chromatin landscape of grain amaranth

Corresponding Author: Professor Markus Stetter

Version 0:

Reviewer comments:

Reviewer #1

(Remarks to the Author)

This study provides the most complete genome assembly, methylome and chromatin map of grain amaranth, and presents a population-scale investigation of chromatin-landscape changes during amaranth domestication. The data resource are valuable for amaranth breeding, and the chromatin changes during domestication is also interesting. But I have some concerns about the key discoveries in this study, as well as the language editing.

As the overall amount of accessible chromatin is highly conserved with only a small fraction differed, so the title "Domestication shaped the chromatin landscape of grain amaranth" may over claimed, or this claim is not very accurate.

In the Results section, the three subtitles "hypochochriacus reference genome and annotation", "The *A. hypochochriacus* chromatin map", "Chromatin changes during amaranth domestication", do not show any meaningful results, they are descriptive but not result-leading. Maybe the authors have performed many analysis in each section, but which is the most important discovery in this study? Or most results are similar to previous studies performed in more popular crops such as rice, maize, and soybean.

Line 52-53: is that 232 scaffolds belong to the 16 chromosomes? What percent of assembled sequences were anchored into the 16 chromosome-level scaffolds?

Line 69: Does "repetitive elements" here mean only TE sequences? Or contain both TE and tandem repeats?

Line 70: "or" should be changed to "and".

Line 139-140: a "," should be added between "species and "an".

Line 188: "is" should be deleted in "This is agrees".

Line 189: "between" should be changed into "among".

Reviewer #2

(Remarks to the Author)

This study presents a comprehensive genomic and epigenomic analysis of grain amaranth domestication, leveraging a high-quality chromosome-level assembly of *A. hypochochriacus* and ATAC-seq data from 18 accessions across five species, including *A. caudatus*, *A. cruentus*, *A. hypochochriacus*, *A. hybridus* and *A. quitensis*. The authors demonstrate that domestication drove species-specific changes in chromatin accessibility, with a bias toward opening chromatin regions. These differentially accessible chromatin regions (dACRs) overlap significantly with selective sweeps, supporting an

adaptive role for chromatin remodeling during independent domestications. This article merely conducts rudimentary analyses and correlation analyses; it fails to delve into the mechanistic insights regarding how domestication drives changes in chromatin accessibility or elucidate its associated impacts. Following are the comments.

1. While dACRs overlapping selective sweeps are compelling, functional links to domestication traits remain correlative. What's effect of dACR on domestication?
2. The observation that domestication preferentially "opens" chromatin is intriguing but mechanistically underexplored. Does increased accessibility correlate with shifts in specific trait modules (e.g., seed size, stress response)?
3. For selective sweep analyses, sample sizes are modest potentially reducing power to detect subtle sweeps.
4. The finding that 78.83% of ACRs overlap TEs and that methylation silences TEs in accessible regions is a highlight. However, the evolutionary significance—e.g., whether TE mobilization in open chromatin contributes to domestication-associated variation—is not addressed. What about the function of DNA methylation in chromatin accessibility changes during domestication.

Reviewer #3

(Remarks to the Author)

In the manuscript, "Domestication shaped the chromatin landscape of grain amaranth", the authors sequenced and assembled the genome of *A. hypochondriacus* and utilized methylation data and ATAC-seq to characterize the population-level chromatin landscape across five *Amaranthus* species, including three domesticated crops and two wild relatives. Their findings reveal chromatin regions in *A. hypochondriacus* that may be associated with selective sweeps and potentially play a role in the species' domestication. While the study provides valuable insights into methylation patterns and chromatin accessibility in grain amaranths, the overall conclusion regarding the proportion of chromatin that changed state is consistent with findings in other species. Please see below for comments.

Ln 9 – 11: The juxtaposition of decreasing genome size and frequent polyploidization in the current phrasing appears contradictory and needs clarification. Consider introducing the concept of genome downsizing following polyploidization to reconcile the two processes.

Ln 30: Not everyone may be familiar with "grain amaranths," so it would be helpful to specify them before beginning the next sentence with "these." Also, consider removing "that" from the phrase "...the Americas that have been domesticated..." for better flow.

Ln 49: "...genomic analyses,"

Ln 58 – 60: There are nuclear fragments that are derived from the organelles, known as NUMTs (nuclear mitochondrial DNA segments) or NUPTs (nuclear plastid DNA segments). The reference cited from Winkler et al. (2024) presented two possible explanations for what might be occurring in Scaffold 10 (Figure S1) in their study: 1) erroneous assembly or 2) nuclear integrants of organellar origin. It is unclear why the current manuscript chose to support the first possibility and then proceeded to interpret the rearrangement in the new assembly as evidence for correction of a misassembled region. On what basis is the region deemed misassembled, especially when Winkler et al. explicitly proposed two possibilities? Moreover, Winkler et al. did not report the lengths of the contigs showing similarity to organellar sequences or the actual percentage identities in their study. We also lack genome-wide data on the insertion of organellar fragments into the nuclear genome for *A. hypochondriacus*. Given these gaps, how does the manuscript confidently rule out NUMTs or NUPTs?

Ln 74 – 75: Comparative analyses among *Amaranthus* species to investigate genome evolution have been conducted recently; see Raiyemo et al. (2025). Chromosome-level assemblies of *Amaranthus palmeri*, *A. retroflexus*, and *A. hybridus* enabled genomic comparisons and the identification of a sex-determining region. Figure 2 in that study closely aligns with Extended Data Figure E2 in the present manuscript. However, that body of work is largely unacknowledged here. Where were the chromosome-level assemblies of *Amaranthus* species used in this study obtained and why were their sources not cited? Notably, the fusion of Chr15 and Chr12 in *A. palmeri* into Chr1 in *A. tuberculatus* is also visible in Extended Data Figure E2, mirroring the observation in Figure 2 of Raiyemo et al. (2025). Please provide appropriate references and discuss the similarities or differences between this study and previous work.

Ln 85: While I agree that most accessible chromatin regions from both leaf and seedling samples fall between 0 and 1,000 bp, what are your thoughts on regions between 1,000 and 3,000 bp? Could these represent super-enhancers, broad highly transcribed gene bodies, or other regulatory elements? Additionally, the choice of a 4,000 bp cutoff to define false positives seems arbitrary. Such claims should be supported with references, and possible reasons for unusually long accessible regions such as peak calling artifacts, poor read alignment, or PCR amplification bias should be discussed. For any regions exceeding 4,000 bp inspected in IGV, is the signal strength consistently high across the entire length?

Ln 87: So only two ACRs were greater than 4,000 bp?

Ln 88: Put a space between 11,238,400 and bp. Same as Ln 93.

Ln 96: This paragraph began by referencing Fig. 1B, but I did not see a reference to Fig. 1A prior to this paragraph or a reference to Fig. 1A anywhere else in the manuscript.

Ln 108: What is a 'mixed' seeding sample?

Ln 109 – 111: I think it is important to relate this sentence to TE composition and activity rather than focusing solely on genome size. Two genomes can have similar sizes but very different TE landscapes including differences in TE types, abundance, and activity which in turn affect methylation patterns differently. Some species may have large genomes with fewer active TEs or employ different TE silencing mechanisms.

Ln 116: The sentence “The increased accessibility of genes, particularly around the TSS, potentially leads to increased expression of genes with open chromatin upstream” is confusing. Consider rephrasing it as: “Increased chromatin accessibility upstream of the TSS is often associated with higher expression of the corresponding gene.”

Ln 119: Here and the methods section (Ln 630 – 637), there is no mention of the number of gene sets within or outside ACRs. How many “open” genes were expression counts obtained for? What is meant by the “equal number” of random genes not associated with ACRs? There is also no information on whether log-transformation was applied to the quantification or counts from Kallisto, leaving readers to infer this from Figure 2 or Figure S8. Also, raw or linear TPM values from Kallisto can be zero. Since the logarithm of 0 is undefined, was any offset added to TPM values before log transformation (commonly $\log_2(\text{TPM} + 1)$ or, in this case $\log_{10}(\text{TPM} + 1)$)? Furthermore, Figure S8 indicates that ANOVA was performed separately for each tissue. It is standard practice to apply a multiple testing correction method (e.g., Bonferroni or a less conservative Benjamini-Hochberg FDR) when conducting multiple hypothesis tests. No such correction appears to have been applied in this analysis. Presenting both raw and adjusted p-values in the figures or supplementary tables would help readers assess both the uncorrected biological trend and the statistical confidence.

Ln 131 – 133: The sentence is worded in such a way that describes gene bodies rather than the TE families. I recommend rewording the sentence. For example, “This enrichment may result from the preference of these TE families to insert near gene bodies, which positions them within open chromatin and prevents their effective silencing through containment in closed chromatin.”

Ln 139: Put a comma between “species” and “an”

Ln 153 – 154: This is quite noteworthy also considering the close relationship among the grain amaranths. I would think a reference bias would be a problem if species from the *Albersia* subgenus were used.

Ln 157: There is no need to capitalize “Principal Component Analysis” here.

Ln 188: Remove “is” in “This is agrees with the...”

Ln 193 – 194: Rewrite as “Of these, we found six that opened and nine that closed in *A. caudatus*, six that opened and eleven that closed in *A. cruentus*, and seven that opened and seven that closed in *A. hypochondriacus*.”

Ln 206: I am not sure what this phrase “...and the *Amaranthus* genus” means here. Remove from the sentence.

Ln 231: “...is particularly pronounced...”

Ln 499: “...standard manufacturer’s protocol...”

Ln 510: “...analyses.”

Ln 512: The reference for Inspector is Chen et al. (2021). The reference for YaHS was provided.

Ln 555: As indicated previously, the source of these genomes should be cited. They were not generated in this study.

Ln 564: The synonymous substitution rate for Amaranthaceae has been a subject of debate. To my knowledge, there are no published reports of a synonymous substitution rate specific to the Amaranthaceae family. In fact, the only reported rate ($0.28 - 0.41 \times 10^{-9}$) was for the *rbcl* gene by Kadereit et al. (2003). This rate was based on fossil calibration and used to date the evolution of C4 photosynthesis in Chenopodiaceae (now Chenopodioideae). The molecular clock presented in Wang et al. was constructed from RelTime in MEGA, where internal nodes were assigned calibration times obtained from TimeTree. The split between *C. quinoa* and *F. tataricum* at 75-100 MYA was used to date their tree. Since RelTime does not use a synonymous substitution rate in its molecular clock like MCMCtree or BEAST, where is the synonymous substitution rate for Amaranthaceae ($9.6E-9$) attributed to Wang et al. derived from? Also, Figure E1B is very similar to Figure 3 in Wang et al., yet the manuscript does not discuss the similarities or differences between the two studies. Finally, what do the red polygons in Figure E1B represent?

Ln 564: Italicize *Amaranthus* species...

In Figure E2, what do the arrows within chromosomes (e.g., chromosomes 10, 5, and 6 in *A. tricolor*) represent?

In Figure E3, write as “megabase” pairs (Mbp).

In Figure S1 and throughout the manuscript, decide on kb or Kbp, and keep consistent.

In Figure S4, the chromosome position in megabase pairs (MB) is not overlapping due to the optimal font size. In Figure E3,

the font size appears to be too large and chromosome position labels are all overlapping.

Provide high-resolution figure in Figure S7. It is blurry as it is and largely not visible.

Version 1:

Reviewer comments:

Reviewer #1

(Remarks to the Author)

All my concerns have been resolved.

Reviewer #2

(Remarks to the Author)

All concerns have been addressed. I have no more comments.

Reviewer #3

(Remarks to the Author)

I appreciate the authors' efforts in carefully addressing my previous concerns, which have been satisfactorily resolved. Upon further inspection of the manuscript, only very minor edits remain; these do not require additional review once addressed. See my comments below.

Ln 12: change to "extent"

Ln 79: change "calculated" to "inferred"

Figure 1D: Some of the TE superfamilies/families are redundant, which stems from the outputs of different programs. EDTA is built on top of RepeatModeler/RepeatMasker outputs but follows the naming convention proposed by Wicker et al. (2007). For example, in the Perl script (rename_RM_TE.pl) located in the "bin" folder of EDTA, the authors of EDTA rename the output from RepeatModeler: entries such as CACTA or EnSpm_CACTA are labeled as DNA/DTH; members of the hAT superfamily, regardless of family, are labeled as DNA/hAT; RC-Helitron or Helitron are labeled as DNA/Helitron; and MULE-MuDR is labeled as DNA/DTM. As currently presented, the figure is difficult to interpret. It shows that ~3% of MULE-MuDR in ACRs is depleted, while ~17% of DNA/DTM in ACRs is enriched, yet MULE-MuDR and DNA/DTM are the same element. Similarly, ~3% of RC-Helitron is depleted in ACRs, while ~37% of Helitron is enriched, yet RC-Helitron and Helitron are the same element. The figure, along with the corresponding results and discussion (lines 143–144), should be reviewed for consistency.

Ln 141: put a space between "1" and "bp"

Ln 672 – 677: Reword and break the sentences. "We compared" was repeated twice.

Figure S10: The figure legend should read "False discovery rate" and not "False discory rate"

Reviewer: 1

This study provides the most complete genome assembly, methylome and chromatin map of grain amaranth, and presents a population-scale investigation of chromatin-landscape changes during amaranth domestication. The data resource are valuable for amaranth breeding, and the chromatin changes during domestication is also interesting. But I have some concerns about the key discoveries in this study, as well as the language editing.

Thank you for your kind assessment and pointing out the importance of study. We have addressed all comments below.

1- As the overall amount of accessible chromatin is highly conserved with only a small fraction differed, so the title “Domestication shaped the chromatin landscape of grain amaranth” may over claimed, or this claim is not very accurate.

It is important to us to make a distinction between the total amount of open chromatin (which indeed is not different and would also not be expected) and the chromatin landscape, which is the fine scale changes. Here we show a clear change in all three domestications with an increased accessibility of the chromatin during domestication and an overlap with selection regions. Hence we think the statement that "domestication shaped the chromatin landscape" is an accurate description of our findings.

2- In the Results section, the three subtitles “hypochochriacus reference genome and annotation”, “The A. hypochochriacus chromatin map”, “Chromatin changes during amaranth domestication”, do not show any meaningful results, they are descriptive but not result-leading. Maybe the authors have performed many analysis in each section, but which is the most important discovery in this study? Or most results are similar to previous studies performed in more popular crops such as rice, maize, and soybean.

Thank you for the suggestion. We have adapted the subtitles to highlight the results better.

We also edited the text to better highlight our advances. We think we make a clear advance compared to previous studies by adding population-scale sample and treating chromatin as a variant type to do selection tests. Some of the highlights of our study are: 1. TEs in open chromatin are silenced via methylation 2. Chromatin is locally reshaped during domestication 3. Chromatin change is biased towards opening, helping to explain gene expression pattern across crops 4. Chromatin selection patterns overlap with genetic selection signals, showing the interplay between the genome, chromatin and selection Some findings overlap with previous findings in 'popular crops'. Findings made once are new and interesting, but found across species they create patterns and prove the more general importance.

3- Line 52-53: is that 232 scaffolds belong to the 16 chromosomes? What percent of assembled sequences were anchored into the 16 chromosome-level scaffolds?

There are a total of 232 scaffolds in the assembly, including the 16 chromosome-level scaffolds. 96.4 % of the sequences were anchored into the 16 chromosome-level scaffolds. We included the information in the main text.

4- Line 69: Does “repetitive elements” here mean only TE sequences? Or contain both TE and tandem repeats?

We have rewritten the sentence to clarify that 50.43 % of the assembly were annotated as TEs and tandem repeats and added a reference to the Supplementary Table detailing the composition.

5- Line 70: “or” should be changed to “and”

resolved

7- Line 139-140: a “,” should be added between “species and “an”.

resolved

8- Line 188: “is” should be deleted in “This is agrees”.

resolved

9- Line 189 “between” should be changed into “among”.

resolved

Reviewer: 2

Comments to the Author This study presents a comprehensive genomic and epigenomic analysis of grain amaranth domestication, leveraging a high-quality chromosome-level assembly of *A. hypochondriacus* and ATAC-seq data from 18 accessions across five species, including *A. caudatus*, *A. cruentus*, *A. hypochondriacus*, *A. hybridus* and *A. quitensis*. The authors demonstrate that domestication drove species-specific changes in chromatin accessibility, with a bias toward opening chromatin regions. These differentially accessible chromatin regions (dACRs) overlap significantly with selective sweeps, supporting an adaptive role for chromatin remodeling during independent domestications. This article merely conducts rudimentary analyses and correlation analyses; it fails to delve into the mechanistic insights regarding how domestication drives changes in chromatin accessibility or elucidate its associated impacts. Following are the comments.

Thank you very much for your assessment and suggestions. We would like to refer to the response of point to by reviewer 1 regarding the importance of our study. We did not aim do explain only a single chromatin change, but rather analyze the pattern of chromatin and its changes during domestication. This becomes possible through our careful sampling and the repeated nature of amaranth domestication. What we show is how widely chromatin changes are distributed and that they produce a polygenic pattern and interplay with selection. Though we have now added one example where the difference in chromatin also shows a difference in phenotype (red tissue pigmentation). We have addressed all comments below.

1- While dACRs overlapping selective sweeps are compelling, functional links to domestication traits remain correlative. What’s effect of dACR on domestication?

While there are multiple single locus large effect mutations that are causal for significant domestication traits the majority of domestication traits (e.g., seed size, flowering time, fruit size, etc.) are highly polygenic across the genome. The polygenic patter of chromatin changes we find fits well with this more recent view on domestication and the (epi-)genetic changes that go along with it.

2- The observation that domestication preferentially “opens” chromatin is intriguing but mechanistically underexplored. Does increased accessibility correlate with shifts in specific trait modules (e.g., seed size, stress response)?

We show that regions that are accessible have higher expression. Furthermore, previous research showed that genes under selection during domestication have an overall higher expression level than their wild ancestors [3] which improved traits such as seed size, seed quantity, etc. Taken together with their association with signs of selection this points towards a role of dACRs influencing traits through increased expression. We are currently running large scale phenotyping experiments to reveal the genetic control of domestication traits in amaranth that can give further insight into the role of chromatin in specific traits. The fact that chromatin changed during domestication makes this a promising avenue to follow. Once we have even more data types, we are sure to return this fascinating question.

3- For selective sweep analyses, sample sizes are modest potentially reducing power to detect subtle sweeps.

We used in total 88 accessions splitting into 28 *A. caudatus*, 21 *A. cruentus*, 18 *A. hypochondriacus*, 9 *A. hybridus* and 12 *A. quitensis* accessions, respectively. This is a well curated sample of accessions, that has been previously used in Stetter et al. [4], Gonçalves-Dias and Stetter [1] and Gonçalves-Dias et al. [2] to identify selective sweeps. While the number of individuals per population is important, other factors have to be balanced. One important point for the sweep analysis is to prevent misclassifications of accessions into the species as this would ‘dilute’ the sweep. Hence, we take great care to prevent this through curating the set of accessions and removed clearly admixed individuals and misclassifications. We work with these samples intensively and are confident that they represent the crop species and their wild relatives well.

4- The finding that 78.83% of ACRs overlap TEs and that methylation silences TEs in accessible regions is a highlight. However, the evolutionary significance—e.g., whether TE mobilization in open chromatin

contributes to domestication-associated variation—is not addressed. What about the function of DNA methylation in chromatin accessibility changes during domestication.

Thank you for pointing out the importance of our finding. We do see a clear depletion in methylation at TSS and where chromatin is open (Fig. 2a), showing the interplay between chromatin and methylation. From the literature, the role of methylation in TE silencing is well-described. We add here the nuance that the interplay between chromatin and methylation might improve the understanding of the pattern. What we can show here is that methylation related-silencing is stronger (more important) in ACRs than outside (where TEs are probably silenced already by closed chromatin). We observe an increase of methylation near the edges of TEs followed by steep drop if bordering an ACR (1), showing the relationship between ACRs, methylation and TEs.

We did not further study the direct interaction between methylation and chromatin opening during domestication as we produced a methylation map only for the reference individual not the full set of accessions. This would be an interesting further step that we would like to follow in the future.

Figure 1: **A.** Methylation density along the relative position within ACRs, calculated across all ACRs which overlapped with a TE. **B.** TE density along the relative position within ACRs, calculated across all ACRs which overlapped with a TE.

Reviewer: 3

Comments to the Author In the manuscript, “Domestication shaped the chromatin landscape of grain amaranth”, the authors sequenced and assembled the genome of *A. hypochondriacus* and utilized methylation data and ATAC-seq to characterize the population-level chromatin landscape across five *Amaranthus* species, including three domesticated crops and two wild relatives. Their findings reveal chromatin regions in *A. hypochondriacus* that may be associated with selective sweeps and potentially play a role in the species’ domestication. While the study provides valuable insights into methylation patterns and chromatin accessibility in grain amaranths, the overall conclusion regarding the proportion of chromatin that changed state is consistent with findings in other species. Please see below for comments.

Thank you for your feedback and particularly for your detailed comments and edit suggestions that we incorporated. They clarified and improved the manuscript.

8- Ln 9 – 11: The juxtaposition of decreasing genome size and frequent polyploidization in the current phrasing appears contradictory and needs clarification. Consider introducing the concept of genome downsizing following polyploidization to reconcile the two processes.

Thank you for pointing out the unclear nature of this. we rewrote the paragraph to remove the focus on the macro-evolution of genomes (including polyploidy) to focus more on the population genetic signals.

9- Ln 30: Not everyone may be familiar with “grain amaranths,” so it would be helpful to specify them before beginning the next sentence with “these.” Also, consider removing “that” from the phrase “. . . the Americas that have been domesticated. . .” for better flow.

resolved

10- Ln 49: “. . . genomic analyses,”

resolved

11- Ln 58 – 60: There are nuclear fragments that are derived from the organelles, known as NUMTs (nuclear mitochondrial DNA segments) or NUPTs (nuclear plastid DNA segments). The reference cited from Winkler et al. (2024) presented two possible explanations for what might be occurring in Scaffold 10 (Figure S1) in their study: 1) erroneous assembly or 2) nuclear integrants of organellar origin. It is unclear why the current manuscript chose to support the first possibility and then proceeded to interpret the rearrangement in the new assembly as evidence for correction of a misassembled region. On what basis is the region deemed misassembled, especially when Winkler et al. explicitly proposed two possibilities? Moreover, Winkler et al. did not report the lengths of the contigs showing similarity to organellar sequences or the actual percentage identities in their study. We also lack genome-wide data on the insertion of organellar fragments into the nuclear genome for *A. hypochondriacus*. Given these gaps, how does the manuscript confidently rule out NUMTs or NUPTs?

We have clarified our assessment of misassembly in the reference genome v2.2 compared to the genome assembled in this study regarding both chromosome 10, which included regions with high similarity to organellar sequences, and chromosome 11, which showed a large inversion between the assemblies. This assessment is primarily based on the position of contig borders in the assembly, for which we added a visualization in Supplementary Figures S3 and S4. In the assembly v3, chromosome 10 consists of a single contig from ~3 Mb on, whereas scaffold 10 of the assembly v2.2 consists of many contigs which often co-localize with breaks in synteny. While genomic differences between individual plants are possible, breaks in inferred synteny between the assemblies are more likely to be indicative of technical differences since both genome assemblies are based on the same inbred accession ‘Plainsman’. Absence of the region with organellar similarity in the single contig spanning the region of chromosome 10 in assembly v3 show the absence of organellar fragments in chromosome 10 of ‘Plainsman’.

Chromosome 11 in the assembly v3 featured a large inversion compared to scaffold 11 of the assembly v2.2. In comparison to v2.2, the assembled chromosome 11 in v3 consists of a single contig from ~8 Mb on, which spans the right border of the apparent inversion. Like for chromosome 10, for chromosome 11 co-localization of contig breaks with the border of the apparent inversion in v2.2 and the assembly in this area from a single contig in v3 demonstrate the correction a misassembled region. We have adjusted the text in the main paper to reflect our assessment of misassembly correction and refer to the two supplementary figures.

- 12 Ln 74 – 75: Comparative analyses among *Amaranthus* species to investigate genome evolution have been conducted recently; see Raiyemo et al. (2025). Chromosome-level assemblies of *Amaranthus palmeri*, *A.*

retroflexus, and *A. hybridus* enabled genomic comparisons and the identification of a sex-determining region. Figure 2 in that study closely aligns with Extended Data Figure E2 in the present manuscript. However, that body of work is largely unacknowledged here. Where were the chromosome-level assemblies of *Amaranthus* species used in this study obtained and why were their sources not cited? Notably, the fusion of Chr15 and Chr12 in *A. palmeri* into Chr1 in *A. tuberculatus* is also visible in Extended Data Figure E2, mirroring the observation in Figure 2 of Raiyemo et al. (2025). Please provide appropriate references and discuss the similarities or differences between this study and previous work.

Thank you for pointing the relevant citation and comparison. Sorry for overlooking this before. We have included it in our study as suggested. All sources for resources that were used in this study have been added to the method section.

- 13 Ln 85: While I agree that most accessible chromatin regions from both leaf and seedling samples fall between 0 and 1,000 bp, what are your thoughts on regions between 1,000 and 3,000 bp? Could these represent super-enhancers, broad highly transcribed gene bodies, or other regulatory elements? Additionally, the choice of a 4,000 bp cutoff to define false positives seems arbitrary. Such claims should be supported with references, and possible reasons for unusually long accessible regions such as peak calling artifacts, poor read alignment, or PCR amplification bias should be discussed. For any regions exceeding 4,000 bp inspected in IGV, is the signal strength consistently high across the entire length?

The ACRs of 1000-3000bp length are potentially regulatory elements as they show a stronger association with gene bodies than smaller ACRs (77.22% of large ACRs compared to 71.63%). The aim of the upper bound is to limit potential artifacts. We only found 2 ACRs larger than 4,000 bp before joining overlapping ACRs, hence they only contribute little to the overall pattern of ACRs.

- 14 Ln 87: So only two ACRs were greater than 4,000 bp?

Correct. We now clarify the number of excluded ACRs in the text.

- 15 Ln 88: Put a space between 11,238,400 and bp. Same as Ln 93.

resolved

- 16 Ln 96: This paragraph began by referencing Fig. 1B, but I did not see a reference to Fig. 1A prior to this paragraph or a reference to Fig. 1A anywhere else in the manuscript.

now referenced in line 84

- 17 Ln 108: What is a 'mixed' seeding sample?

corrected the spelling mistake "seeding" to "seedling". 'mixed' in this context referred to the fact that seedlings are not one homogeneous tissue but comprised of several specialized tissues (i.e., cotyledons, hypocotyl, roots). We have clarified our meaning in the corresponding section.

- 18 Ln 109 – 111: I think it is important to relate this sentence to TE composition and activity rather than focusing solely on genome size. Two genomes can have similar sizes but very different TE landscapes including differences in TE types, abundance, and activity which in turn affect methylation patterns differently. Some species may have large genomes with fewer active TEs or employ different TE silencing mechanisms.

We fully agree with this assessment. However they do empirically correlate to some degree, hence we wanted to give some guidance from species with different genome sizes to readers who are less familiar with the amaranth genome.

- 19 Ln 116: The sentence "The increased accessibility of genes, particularly around the TSS, potentially leads to increased expression of genes with open chromatin upstream" is confusing. Consider rephrasing it as: "Increased chromatin accessibility upstream of the TSS is often associated with higher expression of the corresponding gene."

We have rephrased the sentence according to the suggestion.

- 20 Ln 119: Here and the methods section (Ln 630 – 637), there is no mention of the number of gene sets within or outside ACRs. How many "open" genes were expression counts obtained for? What is meant by the "equal number" of random genes not associated with ACRs? There is also no information on whether log-transformation was applied to the quantification or counts from Kallisto, leaving readers to infer this from Figure 2 or Figure S8. Also, raw or linear TPM values from Kallisto can be zero. Since the logarithm of 0 is undefined, was any offset added to TPM values before log transformation (commonly $\log_2(\text{TPM} + 1)$ or, in this case $\log_{10}(\text{TPM} + 1)$)? Furthermore, Figure S8 indicates that ANOVA was performed separately

for each tissue. It is standard practice to apply a multiple testing correction method (e.g., Bonferroni or a less conservative Benjamini-Hochberg FDR) when conducting multiple hypothesis tests. No such correction appears to have been applied in this analysis. Presenting both raw and adjusted p-values in the figures or supplementary tables would help readers assess both the uncorrected biological trend and the statistical confidence.

Thank you for your suggested improvements to our methods section. A total of 10,473 genes were associated with ACRs (ACR within 2000 bp of TSS) and could be assigned expression data. As the number of closed genes (no ACR within 2000 bp of TSS) was larger than accessible genes, we randomly selected 10,473 to prevent biases due to different set sizes. We transformed the expression data using $\log_{10}(\text{TPM}+0.001)$ and Tukey's HSD was applied for multiple testing correction. We expanded our method section for this analysis and added additional information.

- 21 Ln 131 – 133: The sentence is worded in such a way that describes gene bodies rather than the TE families. I recommend rewording the sentence. For example, “This enrichment may result from the preference of these TE families to insert near gene bodies, which positions them within open chromatin and prevents their effective silencing through containment in closed chromatin.”

We have followed the recommendation and adjusted the sentence accordingly.

- 22 Ln 139: Put a comma between “species” and “an”

resolved

- 23 Ln 153 – 154: This is quite noteworthy also considering the close relationship among the grain amaranths. I would think a reference bias would be a problem if species from the *Albersia* subgenus were used.

Indeed, mapping sequencing reads between grain amaranths works very well (usually a mapping rate higher than 90%). Mapping to more distant species, e.g., from the *Albersia* subgenus and comparing within and between subgenera would likely lead to difficulties. Hence, the assessment of reference bias can help understand potential pitfalls.

- 24 Ln 157: There is no need to capitalize “Principal Component Analysis” here.

resolved

- 25 Ln 188: Remove “is” in “This is agrees with the . . .”

resolved

- 26 Ln 193 – 194: Rewrite as “Of these, we found six that opened and nine that closed in *A. caudatus*, six that opened and eleven that closed in *A. cruentus*, and seven that opened and seven that closed in *A. hypochondriacus*.”

Thank you for the suggested improvement. We have adjusted the text accordingly.

- 27 Ln 206: I am not sure what this phrase “. . . and the *Amaranthus* genus” means here. Remove from the sentence.

We agree that the old placement of the phrase could imply that we provided genome assemblies and methylome data for multiple members of the *Amaranthus* genus. We moved the phrase so our original intend of providing resources that can enable further studies in this genus becomes more clear.

- 28 Ln 231: “. . . is particularly pronounced. . .”

resolved

- 29 Ln 499: “. . . standard manufacturer’s protocol. . .”

resolved

- 30 Ln 510: “. . . analyses.”

resolved

- 31 Ln 512: The reference for Inspector is Chen et al. (2021). The reference for YaHS was provided.

Thank you for recognizing the mix-up in citation. We added the Chen et al. citation.

- 32 Ln 555: As indicated previously, the source of these genomes should be cited. They were not generated in this study.

All citations for resources not generated in this study have been added.

- 33 Ln 564: The synonymous substitution rate for Amaranthaceae has been a subject of debate. To my knowledge, there are no published reports of a synonymous substitution rate specific to the Amaranthaceae family. In fact, the only reported rate ($0.28 - 0.41 \times 10^{-9}$) was for the *rbcL* gene by Kadereit et al. (2003). This rate was based on fossil calibration and used to date the evolution of C4 photosynthesis in Chenopodiaceae (now Chenopodioideae). The molecular clock presented in Wang et al. was constructed from RelTime in MEGA, where internal nodes were assigned calibration times obtained from TimeTree. The split between *C. quinoa* and *F. tataricum* at 75-100 MYA was used to date their tree. Since RelTime does not use a synonymous substitution rate in its molecular clock like MCMCtree or BEAST, where is the synonymous substitution rate for Amaranthaceae ($9.6E-9$) attributed to Wang et al. derived from? Also, Figure E1B is very similar to Figure 3 in Wang et al., yet the manuscript does not discuss the similarities or differences between the two studies.

Thank you for the careful assessment regarding the substitution rate used in our analysis. The rate of $9.6E-9$ substitutions/site/year was not directly reported by Wang et al. (2024) but was derived from their data as follows: Wang et al. reported a synonymous divergence (*Ks* peak) of 0.63 between *Amaranthus tricolor* and *Beta vulgaris*, and dated this divergence to approximately 32.81 Mya. Using the standard synonymous substitution-based molecular clock formula (Divergence time (Mya) = $[Ks/(2 * \text{Substitution rate})] * 10^6$) we estimated the rate as $9.6E-9$ (i.e., $0.63/(2 * 32.81 * 10^6) = 9.6E-9$). This approach is commonly used when lineage-specific rates are unavailable, but we acknowledge its limitations—including assumptions of rate constancy across lineages and genes. To address uncertainty, we have provided confidence intervals for our WGD time estimates. Our Figure E1B does resemble Figure 3 in Wang et al., and we now include a discussion comparing the two in the revised manuscript to clarify methodological similarities and differences.”

Finally, what do the red polygons in Figure E1B represent?

The red diamonds in Figure E1B indicate estimated points of WGD inferred from peaks in *Ks* in Figure E1A. We have adjusted the Figure description

- 34 Ln 564: Italicize *Amaranthus* species. . .

Fixed

- 35 In Figure E2, what do the arrows within chromosomes (e.g., chromosomes 10, 5, and 6 in *A. tricolor*) represent?

In Figure E2 the arrows within chromosomes indicate that the chromosome in question was reverse complemented in that Figure. We have adjusted the Figure description.

- 36 In Figure E3, write as “megabase” pairs (Mbp).

Corrected

- 37 In Figure S1 and throughout the manuscript, decide on kb or Kbp, and keep consistent.

all mentions of Kbp were changed to kb

- 38 In Figure S4, the chromosome position in megabase pairs (Mb) is not overlapping due to the optimal font size. In Figure E3, the font size appears to be too large and chromosome position labels are all overlapping.

Corrected

- 39 Provide high-resolution figure in Figure S7. It is blurry as it is and largely not visible.

Corrected

References

[1] José Gonçalves-Dias and Markus G Stetter. Popamaranth: a population genetic genome browser for grain amaranths and their wild relatives. *G3*, 11(7):jkab103, 2021.

- [2] José Gonçalves-Dias, Akanksha Singh, Corbinian Graf, and Markus G Stetter. Genetic incompatibilities and evolutionary rescue by wild relatives shaped grain amaranth domestication. *Molecular Biology and Evolution*, 40(8):msad177, 2023.
- [3] Zachary H Lemmon, Robert Bukowski, Qi Sun, and John F Doebley. The role of cis regulatory evolution in maize domestication. *PLoS genetics*, 10(11):e1004745, 2014.
- [4] Markus G Stetter, Mireia Vidal-Villarejo, and Karl J Schmid. Parallel seed color adaptation during multiple domestication attempts of an ancient new world grain. *Molecular Biology and Evolution*, 37(5):1407–1419, 2020.

Reviewer 1

All my concerns have been resolved.

Reviewer 2

All concerns have been addressed. I have no more comments.

Reviewer 3

I appreciate the authors' efforts in carefully addressing my previous concerns, which have been satisfactorily resolved. Upon further inspection of the manuscript, only very minor edits remain; these do not require additional review once addressed. See my comments below.

Thank you for your detailed feedback. We have incorporated all the additional suggestions that helped us improving the manuscript.

- 1 Ln 12: change to “extent” corrected
- 2 Ln 79: change “calculated” to “inferred” corrected

- 3 Figure 1D: Some of the TE superfamilies/families are redundant, which stems from the outputs of different programs. EDTA is built on top of RepeatModeler/RepeatMasker outputs but follows the naming convention proposed by Wicker et al. (2007).

For example, in the Perl script (rename_RM_TE.pl) located in the "bin" folder of EDTA, the authors of EDTA rename the output from RepeatModeler: entries such as CACTA or EnSpm_CACTA are labeled as DNA/DTH; members of the hAT superfamily, regardless of family, are labeled as DNA/hAT; RC-Helitron or Helitron are labeled as DNA/Helitron; and MULE-MuDR is labeled as DNA/DTM. As currently presented, the figure is difficult to interpret. It shows that 3% of MULE-MuDR in ACRs is depleted, while 17% of DNA/DTM in ACRs is enriched, yet MULE-MuDR and DNA/DTM are the same element. Similarly, 3% of RC-Helitron is depleted in ACRs, while 37% of Helitron is enriched, yet RC-Helitron and Helitron are the same element. The figure, along with the corresponding results and discussion (lines 143–144), should be reviewed for consistency.

Thank you for pointing out this oversight. Groups had been taken from the classifications column from EDTAs output file to give a more complete overview of the TE landscape of ACRs, as it also differentiated between full TEs and MITEs. We have now changed to the more commonly used superfamilies. The results and discussion have been adjusted accordingly, thou our conclusion remains the same.

- 4 Ln 141: put a space between “1” and “bp” corrected
- 5 Ln 672 – 677: Reword and break the sentences. “We compared” was repeated twice. corrected
- 6 Figure S10: The figure legend should read “False discovery rate” and not “False discory rate” corrected